# Half-Sandwich Nickelacarboranes Derived from [7-(MeO(CH₂)₂S)-7,8-C₂B₉H₁₁]⁻

Dmitriy K. Semyonov [1,2], Marina Yu. Stogniy [1,2,*], Kyrill Yu. Suponitsky [3] and Igor B. Sivaev [1,4]

1    A. N. Nesmeyanov Institute of Organoelement Compounds, Russian Academy of Sciences, 28 Vavilov Str., 119334 Moscow, Russia

2    M. V. Lomonosov Institute of Fine Chemical Technology, MIREA—Russian Technological University, 86 Vernadsky Av., 119571 Moscow, Russia

3    N. S. Kurnakov Institute of General and Inorganic Chemistry, Russian Academy of Sciences, 31 Leninskii Av., 119991 Moscow, Russia

4    Basic Department of Chemistry of Innovative Materials and Technologies, G.V. Plekhanov Russian University of Economics, 36 Stremyannyi Line, 117997 Moscow, Russia

*    Correspondence: stogniy@ineos.ac.ru

**Abstract:** New carboranyl thioethers $1\text{-MeO(CH}_2)_n\text{S-1,2-C}_2\text{B}_{10}\text{H}_{11}$ (n = 2, 3) were prepared by the alkylation of the trimethylammonium salt of 1-mercapto-*ortho*-carborane with 1-bromo-2-methoxyethane and 1-bromo-3-methoxypropane, respectively. Their deboronation with cesium fluoride in ethanol gave the corresponding *nido*-carboranes $\text{Cs[7-MeO(CH}_2)_n\text{S-7,8-C}_2\text{B}_9\text{H}_{11}]$ (n = 2, 3). The reactions of $\text{Cs[7-MeO(CH}_2)_2\text{S-7,8-C}_2\text{B}_9\text{H}_{11}]$ with various nickel(II) phosphine complexes $[(\text{dppe})\text{NiCl}_2]$ and $[(\text{R'R}_2\text{P})_2\text{NiCl}_2]$ (R = R' = Ph, Bu; R = Me, R' = Ph; R = Ph, R' = Me, Et) were studied and a series of nickelacarboranes $3,3\text{-dppe-1-MeO(CH}_2)_2\text{S-}closo\text{-}3,1,2\text{-NiC}_2\text{B}_9\text{H}_{10}$ and $3,3\text{-}(\text{R'R}_2\text{P})_2\text{-1-MeO(CH}_2)_2\text{S-}closo\text{-}3,1,2\text{-NiC}_2\text{B}_9\text{H}_{10}$ (R = R' = Bu; R = Me, R' = Ph; R = Ph, R' = Me, Et) was prepared. The molecular crystal structure of $3,3\text{-dppe-1-MeO(CH}_2)_2\text{S-}closo\text{-}3,1,2\text{-NiC}_2\text{B}_9\text{H}_{10}$ was determined by single-crystal X-ray diffraction.

**Keywords:** carboranyl thioethers; *nido*-carborane ligands; phosphine ligands; nickelacarboranes; half-sandwich complexes; synthesis; structure

## 1. Introduction

The discovery of the deboronation reaction of *closo*-carborane to *nido*-carborane under the action of nucleophiles became the starting point for the development of metallacarborane chemistry [1,2]. This was facilitated by the unusual chemical and physical properties of the open-face dicarbollide ligand [*nido*-7,8-C₂B₉H₁₁]²⁻ as well as its similarity to the cyclopentadienyl anion [C₅H₅]⁻ [3–6]. To date, a huge variety of sandwich and half-sandwich complexes of transition metals with dicarbollide ligands have been synthesized [7–12]. Many of them have shown excellent prospects for use in a wide variety of fields, from which catalysis [13–16] and medicine [17–28] stand out. At the same time, it should be noted that the degree of knowledge of metallacarboranes varies greatly depending on the complexing metal. The most studied are sandwiched cobalt bis(dicarbollide) complexes [29–31]. The synthesis and properties of half-sandwich complexes of ruthenium, rhodium, and iridium, which are intensively used in catalysis, has also been widely investigated [13,15,16,32–36]. At the same time, the chemistry of nickelacarboranes remains poorly studied and is mainly associated with nickel bis(dicarbollide) complexes [37].

The few examples of half-sandwich nickelacarboranes are represented by nickel(II) complexes both with the parent dicarbollide ligand [7,38–43] and with its *C*- or *B*-substituted derivatives [44–52]. In this contribution, we describe the synthesis of nickel phosphine complexes based on novel carboranyl thioether ligand *nido*-[7-MeO(CH₂)₂S-7,8-C₂B₉H₁₁]⁻

## 2. Results and Discussion

It is well known that the introduction of a substituent into the *nido*-carborane cage can significantly affect the structure and properties of the resulting complexes thereof. For example, the use of *nido*-carborane ligands with the charge-compensating substituent [53] makes it possible to reduce the total charge of the system and stabilize the metal atom in a lower oxidation state [54]. The introduction of substituent(s) with additional donor groups can lead to the formation of complexes in which the metal is coordinated not with the open pentagonal face of *nido*-carborane (classical $\eta^5$-coordination), but with the side substituent (so-called *exo*-complexes) [55–57]. There are also several examples of complexes where *nido*-carborane acts as a $\eta^5:\kappa^1$- or $\eta^5:\kappa^2$-ligand with a side substituent participating in the coordination of the metal center along with the open pentagonal face of the *nido*-carborane basket [47,51,52,58–67]. In addition, the presence of a substituent can cause steric hindrances in the mutual rotation of ligands or even prevent the formation of a complex [68–70].

In our previous work, we began studying the effect of a substituent at the carbon atom in the *nido*-carborane ligand on its ability to form nickel(II) half-sandwich phosphine complexes. The complexation of dicarbollide ligands containing methyl, phenyl, and *N,N'*-dicyclohexylamidine substituents with the participation of nickel phosphine complexes [(Ph$_3$P)$_2$NiCl$_2$], [(MePh$_2$P)$_2$NiCl$_2$], and [(dppe)NiCl$_2$] was studied [71].

In the present study, we prepared two new *C*-substituted *nido*-carboranes with a methoxy group as an additional donor group in the side substituent. For this purpose, the previously developed method of the alkylation of trimethylammonium salt of mercapto-*closo*-carborane [72,73] was applied using 1-bromo-2-methoxyethane and 1-bromo-3- methox ypropane as alkylating agents (Scheme 1). The use of this method made it possible to avoid substitution at the second carbon atom of the *ortho*-carborane cage and led to the preparation of carboranyl thioethers **1** and **2** in two steps with moderate (compound **1**) and good (compound **2**) yields. The deboronation of *ortho*-carborane derivatives **1** and **2** with cesium fluoride gave the corresponding anionic *nido*-carboranes **3** and **4** (Scheme 1).

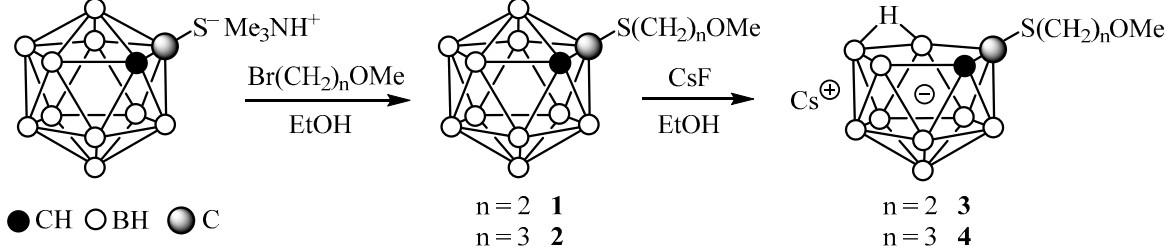

**Scheme 1.** Synthesis of *closo*- and *nido*-carboranyl thioethers **1–4**.

The preparation of compounds **1–4** was confirmed by NMR spectroscopy data. For example, the conversion of *closo*-compounds **1** and **2** into *nido*-species **3** and **4** is clearly evidenced by a change in the spectral pattern (both the range of signals and their number) in the $^{11}$B NMR spectra. Thus, the $^{11}$B NMR spectrum of **1** consists of five groups of doublets with an integral ratio 1:1:4:2:2 in the range of $-(2.1$–$12.5)$ ppm, which corresponds to the range of *closo*-carborane derivatives and indicates the presence of a plane of symmetry in the compound. On the contrary, the $^{11}$B NMR spectrum of *nido*-carborane derivative **3** consists of seven groups of signals with the integral ratio 1:1:1:3:1:1:1 in the range of $-(9.8$–$36.6)$ ppm and this spectral pattern corresponds to the region of *nido*-carborane derivatives and indicates the loss of the symmetry plane by the molecule.

Compound **3** was chosen for studying reactions with nickel(II) phosphine complexes. The reaction of **3** with [(dppe)NiCl$_2$] in dry THF using potassium *tert*-butoxide as a base for the deprotonation of *nido*-carborane does not require heating and proceeds completely within 30 min at room temperature under argon atmosphere (Scheme 2). The resulting half-sandwich complex 3,3-dppe-1-MeO(CH$_2$)$_2$S-*closo*-3,1,2-NiC$_2$B$_9$H$_{10}$ (**5**) was isolated by



column chromatography on silica (eluent $CH_2Cl_2$) as dark green crystals with the yield of 68%.

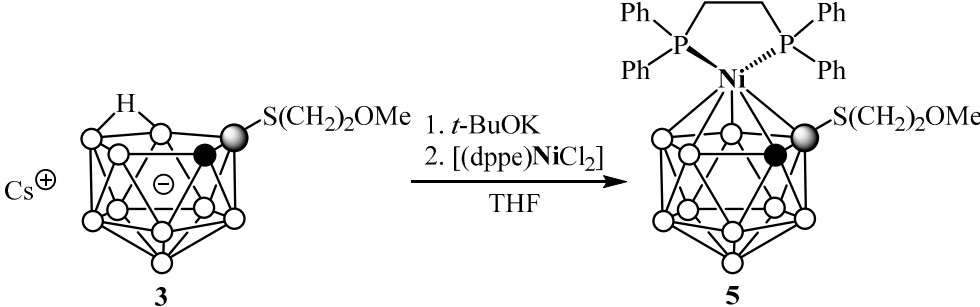

**Scheme 2.** Synthesis of 3,3-dppe-1-MeO$(CH_2)_2$S-*closo*-3,1,2-NiC$_2$B$_9$H$_{10}$ (**5**).

The formation of nickelacarborane **5** is well monitored by $^{11}$B NMR spectroscopy. Figure 1 shows that the signals of the nickel(II) half-sandwich complex **5** are shifted to a lower field relative to the signals of *nido*-carborane **3**. They are in the range from 5.5 to −20.8 ppm and represent a series of six broadened doublets with the integral ratio 1:2:1:2:2:1. The $^1$H NMR spectrum of **5** contains typical multiplets from phosphine ligands in the aromatic region as well as the signals from the dicarbollide side substituent. The methylene protons of the dppe ligand appear as a broad multiplet at 2.87–2.81 ppm. The signal of the unsubstituted C*H* hydrogen of the dicarbollide ligand appears as a broadened singlet at 1.90 ppm. The $^{13}$C NMR spectrum of complex **5** contains a series of doublets from aromatic carbons in the region of 135–128 ppm as a result of their splitting on phosphorus atoms. The $CH_{carb}$ signal appears as a singlet at 52.1 ppm, while the methylene groups of the dppe ligand appear as a doublet at 29.3 ppm with the C–P splitting constant of 23.5 Hz. The $^{31}$P NMR spectrum of **5** contains one singlet at 59.3 ppm.

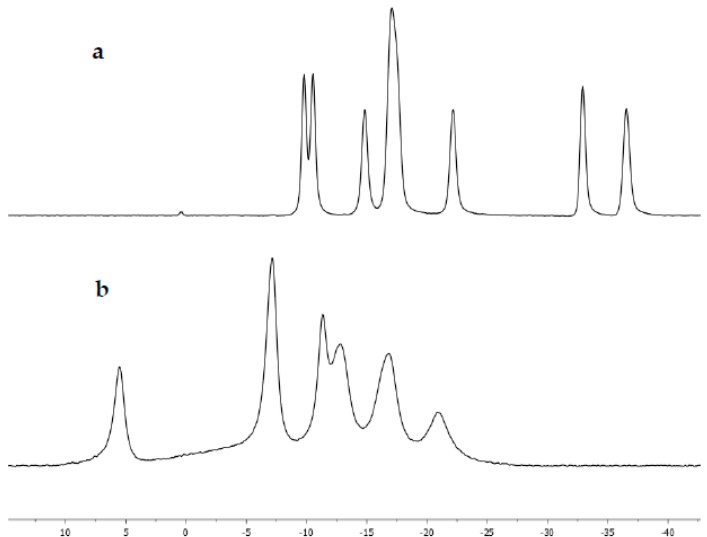

**Figure 1.** The comparison of $^{11}$B{$^1$H} NMR spectra of *nido*-carborane **3** (**a**) and complex **5** (**b**).

The solid state structure of complex **5** was determined by single-crystal X-ray diffraction. An asymmetric unit cell contains two molecules of nickelacarborane 3,3-dppe-1-MeO$(CH_2)_2$S-*closo*-3,1,2-NiC$_2$B$_9$H$_{10}$ and one molecule of acetone solvent. A general view of the nickelacarborane molecule is given in Figure 2. The orientation of the dicarbollide ligand in complex **5**, determined by the dihedral angle between the P1−Ni1−P2 plane and the B8−Ni1−Center(C1−C2) plane, which is 78.9(2)° and 74.5(2)° for two independent molecules, slightly deviates from the nearly ideal electronically controlled orientation found in the structure of 3,3-dppe-*closo*-3,1,2-NiC$_2$B$_9$H$_{11}$ (the similar dihedral angle of ~89°) [43]

and is close to that found in 3,3-dppe-1-Ph-*closo*-3,1,2-NiC$_2$B$_9$H$_{10}$ (the dihedral angle of ~83°) [71].

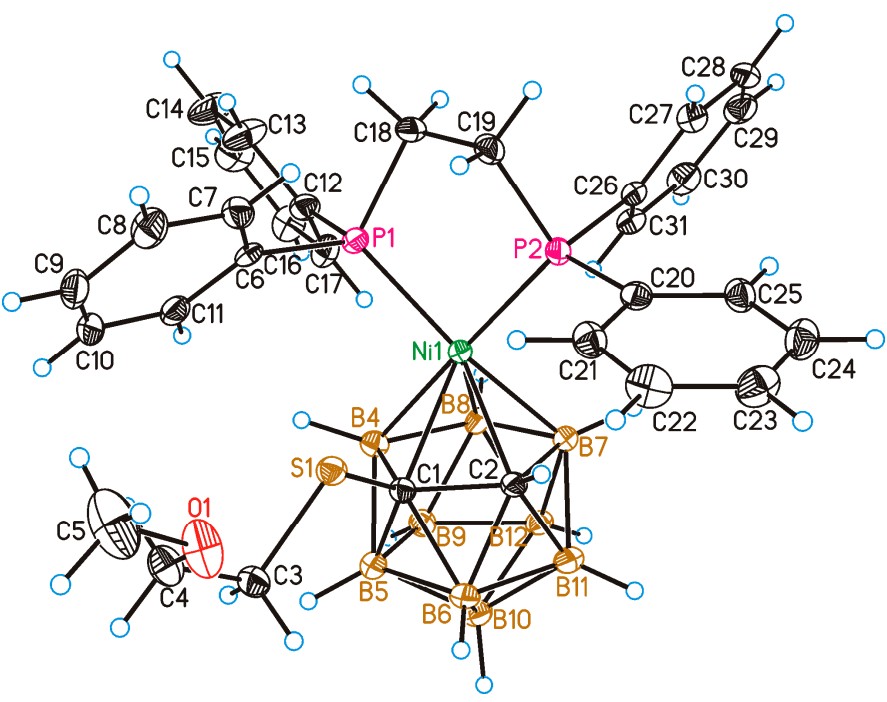

**Figure 2.** General view of the molecule of 3,3-dppe-1-MeO(CH$_2$)$_2$S-*closo*-3,1,2-NiC$_2$B$_9$H$_{10}$ (**5**) showing the numbering scheme. Thermal ellipsoids are given at the 50% probability level.

The two independent molecules in structure **5** adopt a slightly different structure due to the differences in the conformation of the dppe ligand, as shown in Figures 3 and S1.

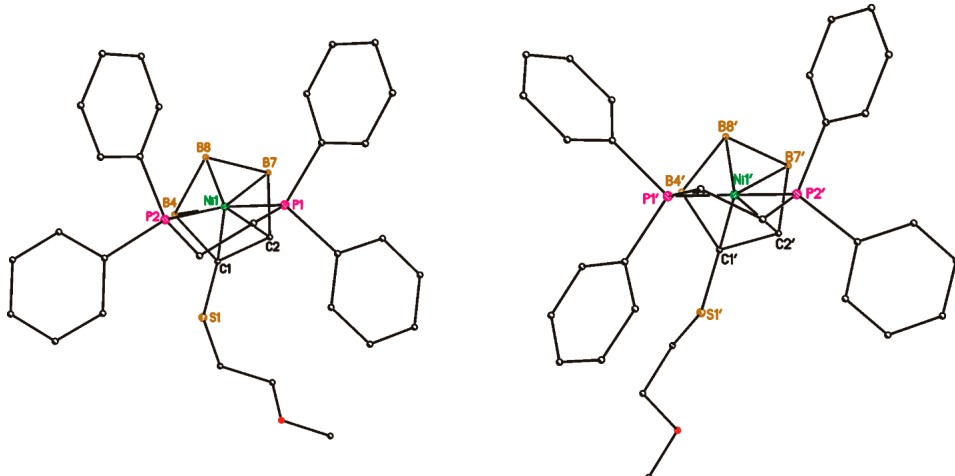

**Figure 3.** Comparison of two symmetrically independent molecules in the structure of complex **5**. A (**left**) and A' (**right**) shown as in the projections onto the open face (C1–C2–B7–B8–B4) of the dicarbollide ligand.

More pronounced differences are observed for the orientation of the SCH$_2$CH$_2$OMe substituent (see Figures 3 and S1), which can be explained by the influence of the crystal packing. This means that symmetry-independent molecules have a somewhat different environment in the unit cell which is clearly represented by their interaction with the solvent acetone. In both molecules, the CH fragment of the phenyl ring of the dppe ligand

is involved in H-bonding with the solvent molecule. However, the interaction of acetone with molecule A (C25-H25A···O1S: C-H, 0.95 Å; O···H, 2.57 Å; C···O, 3.312(3) Å; <CHO, 136°) is weaker than that with A' (C27-H27B···O1S: C-H, 0.95 Å; O···H, 2.40 Å; C···O, 3.268(3) Å; <CHO, 152°) (Figure 4).

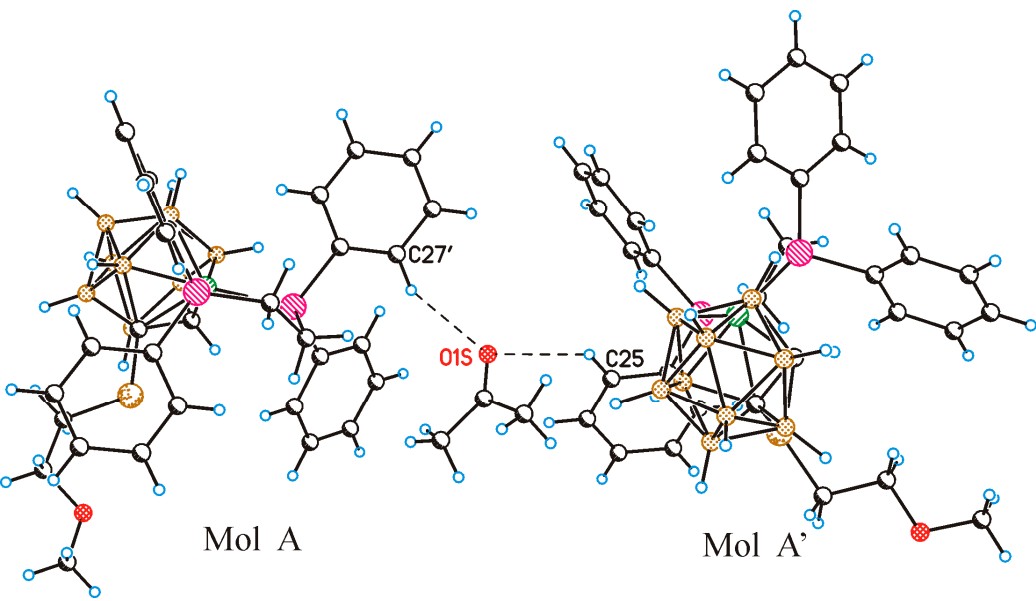

**Figure 4.** Intermolecular hydrogen bonding in the crystal structure of 3,3-dppe-1-MeO(CH$_2$)$_2$S-*closo*-3,1,2-NiC$_2$B$_9$H$_{10}$·0.5Me$_2$CO.

Next, we studied the metalation of *nido*-carborane **3** with different phosphine complexes of nickel(II) [(R′R$_2$P)$_2$NiCl$_2$] (R = R′ = Ph, Bu; R = Me, R′ = Ph; R = Ph, R′ = Me, Et). It was expected that the methoxy group of the side substituent with a pair of electrons on an oxygen atom could participate in the complexation with the metal, as observed earlier [58,60,62], along with the displacement of one phosphine ligand. The reaction conditions were the same as those for the synthesis of complex **5**. However, as a result, a series of half-sandwich nickel(II) complexes with two phosphine ligands (which is without the direct participation of side substituent in the formation of complex) **6–9** was obtained (Scheme 3). The reaction of **3** with [(Ph$_3$P)$_2$NiCl$_2$] did not lead to the formation of any nickelacarborane and only the starting *nido*-carborane was isolated from the reaction mixture.

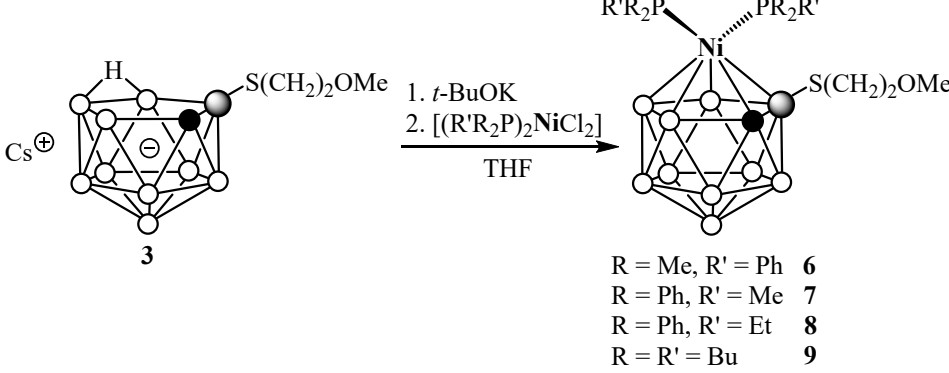

**Scheme 3.** Syntheses of nickelacarboranes **6–9**.

We assume that the absence of metallacarborane product in the reaction of **3** with [(Ph$_3$P)$_2$NiCl$_2$] is due to steric hindrances caused by the presence of bulky triphenylphosphine ligands (the Tolman cone angle $\theta$ is 145°) along with the substituent at carbon atom

in the open pentagonal face of *nido*-carborane. We observed a similar result earlier in an attempt to metalate *C*-substituted *nido*-carboranes with the $[(Ph_3P)_2NiCl_2]$ complex, even in the case of a such small substituent as the methyl group in 7-methyl-*nido*-carborane [71]. However, the replacement of at least one phenyl group by alkyl one (the Tolman cone angles are 140, 136, and 112° for $PPh_2Et$, $PPh_2Me$, and $PPhMe_2$, respectively), as well as the use of a $PBu_3$ ligand (the Tolman cone angle is 132°), leads to the formation of half-sandwich nickelacarboranes as it was supported by the $^{11}B$ NMR spectroscopy data. The signals of complexes **6–9** in the $^{11}B$ NMR spectra appear as a set of five to eight broadened doublets in the characteristic range of $-(3.0–23.0)$ ppm (see Supplementary Materials).

The most clear evidence of the presence of steric hindrance in complexes **6–9** is their $^{31}P$ NMR spectra, each of which contains doublets of two nonequivalent phosphine ligands with the $J$(P–P) splitting constants of ~0 Hz. The $^1H$ and $^{13}C$ NMR spectra of **6–9** also indicate the absence of the free rotation of ligands due to steric hindrances. In the $^1H$ NMR spectra, the signals of the $CH_{carb}$ groups of the dicarbollide ligand are the most indicative. For example, in the $^1H$ NMR spectra of complexes **6** and **7**, these signals appear as high-field doublets with $^4J$(P-H) splitting constants of 10.8 and 14.3 Hz, respectively. The signals of $CH_{carb}$ groups also appear as doublets in the $^{13}C$ NMR spectra of **6–9**. For example, for complex **9**, it is observed at 41.0 ppm with the $^3J$(P-C) splitting constant of 15.0 Hz. It should be noted that such a spectral pattern is quite characteristic of the complexes with restricted rotation of the dicarbollide ligand [68,69,71].

There is also other spectral evidence of the absence of the free rotation of ligands in complexes **6–9** due to steric hindrances. In particular, in the $^1H$ NMR spectrum of complex **7**, the signals of the methyl groups of the $PPh_2Me$ ligands appear as doublets at 1.93 ppm ($J$ = 9.8 Hz) and 1.55 ppm ($J$ = 8.1 Hz) (Figure 5), while in the $^{13}C$ NMR spectrum, the corresponding doublets are located at 20.2 ppm ($J$ = 39.5 Hz) and 10.0 ppm ($J$ = 33.6 Hz).

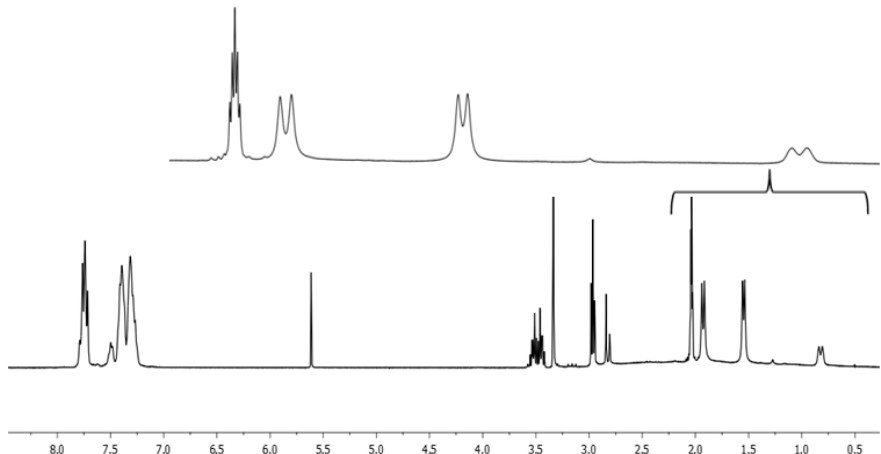

**Figure 5.** The $^1H$ NMR spectrum of complex **7**.

## 3. Conclusions

In this work, the reactions of *C*-substituted *nido*-carboranyl thioether $[7-(MeO(CH_2)_2S)-7,8-C_2B_9H_{11}]^-$ (**3**) with various nickel(II) phosphine complexes were studied. The reaction of ligand **3** with (dppe)NiCl$_2$ was found to result in the formation of half-sandwich complex 3,3-dppe-1-MeO(CH$_2$)$_2$S-*closo*-3,1,2-NiC$_2$B$_9$H$_{10}$ (**5**), the structure of which was determined by single crystal X-ray diffraction. The metallation of **3** with diphosphine nickel complexes $[(R'R_2P)_2NiCl_2]$ (R = R′ = Bu; R = Me, R′ = Ph; R = Ph, R′ = Me, Et) led to the corresponding nickelacarboranes 3,3-(R'R$_2$P)$_2$-1-MeO(CH$_2$)$_2$S-*closo*- 3,1,2-NiC$_2$B$_9$H$_{10}$ (R = R′ = Bu; R = Me, R′ = Ph; R = Ph, R′ = Me, Et) (**6–9**) containing two phosphine ligands. The donor methoxy group in the side chain of the dicarbollide ligand does not participate in the complexation with metal. However, the presence of the side substituent causes steric hindrances that prevent the free rotation of the ligands, which is clearly reflected by the NMR spectroscopy

data. The synthesized nickelacarboranes are of interest as potential catalysts for selective carbene-transfer reactions. Their biological activity will also be investigated.

## 4. Experimental Section

### 4.1. Materials and Methods

Trimethylammonium salt of 1-mercapto-*ortho*-carborane [73], dichloro(1,2-bis(diphenyl phosphino)ethane)nickel(II) [(dppe)NiCl$_2$], dichlorobis(triphenylphosphine)nickel (II) [(Ph$_3$P)$_2$NiCl$_2$], dichlorobis(dimethylphenylphosphine)nickel(II) [(Me$_2$PhP)$_2$NiCl$_2$], dichlorobis(methyldiphenylphosphine)nickel(II) [(MePh$_2$P)$_2$NiCl$_2$], dichlorobis(ethyldiphe nylphosphine)nickel(II) [(EtPh$_2$P)$_2$NiCl$_2$], and dichlorobis(tributhylphosphine)nickel(II) [(Bu$_3$P)$_2$NiCl$_2$] [74] were synthesized according to methods described in the literature. Tetrahydrofuran was dried using the standard procedure [75]. 1-Bromo-2-methoxyethane and 1-bromo-3-methoxypropane were purchased from Acros Organics and ABCR, correspondingly, and used without purification. The reaction progress was monitored by thin-layer chromatography (Merck F254 silica gel on aluminum plates) and visualized using 0.5% PdCl$_2$ in 1% HCl in aq. MeOH (1:10). Acros Organics silica gel (0.060–0.200 mm) was used for column chromatography. The instruments used to characterize the synthesized compounds is given in the Supplementary Materials.

### 4.2. Synthesis of 1-(MeO(CH$_2$)$_2$S)-1,2-C$_2$B$_{10}$H$_{11}$ (1)

1-Bromo-2-methoxyethane (0.40 mL, 4.25 mmol) was added to a solution of the trimethylammonium salt of 1-mercapto-*ortho*-carborane (1.00 g, 4.25 mmol) in ethanol (40 mL) and the mixture was heated under reflux for approximately 5 h. The reaction mixture was allowed to cool to room temperature and evaporated to dryness in vacuum. The obtained product was isolated by column chromatography on silica with dichloromethane as an eluent to give a yellowish oil of **1** (0.39 g, 40% yield). $^1$H NMR (acetone-d$_6$, ppm): $\delta$ 4.78 (1H, s, C$H_{carb}$), 3.58 (2H, t, $J$ = 6.0 Hz, SCH$_2$C$H_2$OCH$_3$), 3.28 (3H, s, OC$H_3$), 3.20 (2H, t, $J$ = 6.0 Hz, SC$H_2$CH$_2$OCH$_3$), and 3.0 ÷ 1.5 (10H, m, B$H$). $^{13}$C NMR (acetone-d$_6$, ppm): $\delta$ 77.9 (C$S_{carb}$), 70.0 (SCH$_2$CH$_2$OCH$_3$), 68.7 (CH$_{carb}$), 57.8 (OCH$_3$), 36.9 (SCH$_2$CH$_2$OCH$_3$). $^{11}$B NMR (acetone-d$_6$, ppm): $\delta$ −2.1 (1B, d, $J$ = 150 Hz), −5.6 (1B, d, $J$ = 150 Hz), −9.5 (4B, d, $J$ = 138 Hz), −12.1 (2B, d, $J$ = 180 Hz), and −12.5 (2B, d, $J$ = 165 Hz). IR (film, cm$^{-1}$): 3048 ($\nu_{C-H}$), 2940 ($\nu_{C-H}$), 2838 ($\nu_{C-H}$), 2610 (br, $\nu_{B-H}$), 2588 (br, $\nu_{B-H}$), 1479, 1455, 1423, 1385, 1291. ESI HRMS: $m/z$ for C$_5$H$_{18}$B$_{10}$OS: calcd. 258.2763 [M+Na]$^+$, obsd. 258.2776 [M+Na]$^+$.

### 4.3. Synthesis of 1-(MeO(CH$_2$)$_3$S)-1,2-C$_2$B$_{10}$H$_{11}$ (2)

The procedure was similar to that described for **1**, using the trimethylammonium salt of 1-mercapto-*ortho*-carborane (0.10 g, 0.43 mmol) in ethanol (10 mL) and 1-bromo-3-methoxypropane (0.05 mL, 0.43 mmol) to give a yellowish oil of **2** (0.08 g, 71% yield). $^1$H NMR (acetone-d$_6$, ppm): $\delta$ 4.77 (1H, s, C$H_{carb}$), 3.39 (2H, t, $J$ = 5.9 Hz, SCH$_2$CH$_2$C$H_2$OCH$_3$), 3.25 (3H, s, OC$H_3$), 3.06 (2H, t, $J$ = 7.4 Hz, SC$H_2$CH$_2$CH$_2$OCH$_3$), 1.82 (2H, m, $J$ = 6.7 Hz, SCH$_2$C$H_2$CH$_2$OCH$_3$), 2.9 ÷ 1.3 (10H, m, B$H$). $^{13}$C NMR (acetone-d$_6$, ppm): $\delta$ 75.7 (C$S_{carb}$), 70.1 (SCH$_2$CH$_2$CH$_2$OCH$_3$), 68.6 (CH$_{carb}$), 57.7 (OCH$_3$), 34.0 (SCH$_2$CH$_2$CH$_2$OCH$_3$), 28.5 (SCH$_2$CH$_2$CH$_2$OCH$_3$). $^{11}$B NMR (acetone-d$_6$, ppm): $\delta$ −2.1 (1B, d, $J$ = 149 Hz), −5.5 (1B, d, $J$ = 148 Hz), −9.5 (4B, d, $J$ = 145 Hz), −12.1 (2B, d, $J$ = 175 Hz), −12.5 (2B, d, $J$ = 159 Hz). IR (film, cm$^{-1}$): 3070 ($\nu_{C-H}$), 2992 ($\nu_{C-H}$), 2933 ($\nu_{C-H}$), 2884 ($\nu_{C-H}$), 2838 ($\nu_{C-H}$), 2599 (br, $\nu_{B-H}$), 1478, 1449, 1386, 1295. ESI HRMS: $m/z$ for C$_6$H$_{20}$B$_{10}$OS: calcd. 249.2198 [M+H]$^+$, obsd. 249.2197 [M+H]$^+$.

### 4.4. Synthesis of Cs[7-(MeO(CH$_2$)$_2$S)-7,8-C$_2$B$_9$H$_{11}$] (3)

Cesium fluoride (0.51 g, 3.36 mmol) was added to a solution of **1** (0.39 g, 1.68 mmol) in ethanol (30 mL) and the reaction mixture was heated under reflux for approximately 16 h (the end of the reaction was determined by TLC in CHCl$_3$) during which a white precipitate formed. The mixture was cooled to room temperature, the solid was filtered off, and the filtrate was evaporated under reduced pressure. Acetone (20 mL) was added to the

residue and the excess of cesium fluoride was filtered off. The solution was evaporated to dryness to give a white crystalline product **3** (0.57 g, 95% yield). $^1$H NMR (acetone-d$_6$, ppm): $\delta$ 3.49 (2H, m, SCH$_2$C*H*$_2$OCH$_3$), 3.29 (3H, s, OC*H*$_3$), 3.00 (1H, m, SC*H*$_2$CH$_2$OCH$_3$), 2.61 (1H, m, SC*H*$_2$CH$_2$OCH$_3$), 1.94 (1H, s, C*H*$_{carb}$), 3.0 $\div$ (−0.4) (9H, m, B*H*), −2.7 (1H, br m, B*H*B). $^{13}$C NMR (acetone-d$_6$, ppm): $\delta$ 72.6 (S*C*H$_2$CH$_2$OCH$_3$), 57.5 (O*C*H$_3$), 53.0 (*C*H$_{carb}$), 35.5 (S*C*H$_2$CH$_2$OCH$_3$). $^{11}$B NMR (acetone-d$_6$, ppm): $\delta$ −9.8 (1B, d, *J* = 115 Hz), −10.5 (1B, d, *J* = 116 Hz), −14.8 (1B, d, *J* = 161 Hz), −17.0 (3B, d, *J* = 133 Hz), −22.2 (1B, d, *J* = 149 Hz), −32.9 (1B, dd, *J* = 139, 32 Hz), −36.6 (1B, d, *J* = 139 Hz). IR (film, cm$^{-1}$): 2987 ($\nu_{C-H}$), 2937 ($\nu_{C-H}$), 2900 ($\nu_{C-H}$), 2838 ($\nu_{C-H}$), 2539 (br, $\nu_{B-H}$), 1476, 1464, 1447, 1383, 1258. ESI HRMS: *m/z* for C$_5$H$_{18}$B$_9$OS: calcd. *m/z* 224.1962 [M]$^-$, obsd. *m/z* 224.1976 [M]$^-$. UV (acetone, nm): $\lambda$ 313.

### 4.5. Synthesis of Cs[7-(MeO(CH$_2$)$_3$S)-7,8-C$_2$B$_9$H$_{11}$] (**4**)

The procedure was similar to that described for **3**, using **2** (0.07 g, 0.29 mmol) in ethanol (10 mL) and cesium fluoride (0.09 g, 0.58 mmol) to give a white crystalline product **4** (0.13 g, 98% yield). $^1$H NMR (acetone-d$_6$, ppm): $\delta$ 3.40 (2H, m, SCH$_2$CH$_2$C*H*$_2$OCH$_3$), 3.24 (3H, s, OC*H*$_3$), 2.86 (1H, m, SC*H*$_2$CH$_2$CH$_2$OCH$_3$), 2.52 (1H, m, SC*H*$_2$CH$_2$CH$_2$OCH$_3$), 1.95 (1H, s, C*H*$_{carb}$), 1.84 (1H, m, SCH$_2$C*H*$_2$CH$_2$OCH$_3$), 1.73 (1H, m, SCH$_2$C*H*$_2$CH$_2$OCH$_3$), 3.1 $\div$ (−0.4) (9H, m, B*H*), −2.7 (1H, br m, B*H*B). $^{13}$C NMR (acetone-d$_6$, ppm): $\delta$ 71.0 (S*C*H$_2$CH$_2$CH$_2$OCH$_3$), 57.6 (O*C*H$_3$), 53.0 (*C*H$_{carb}$), 33.1 (S*C*H$_2$CH$_2$CH$_2$OCH$_3$), 30.7 (S*C*H$_2$CH$_2$CH$_2$OCH$_3$). $^{11}$B NMR (acetone-d$_6$, ppm): $\delta$ −9.9 (1B, d, *J* = 140 Hz), −10.5 (1B, d, *J* = 140 Hz), −14.9 (1B, d, *J* = 158 Hz), −16.9 (2B, d, *J* = 137 Hz), −17.8 (1B, d, *J* = 121 Hz), −22.3 (1B, d, *J* = 150 Hz). −33.0 (1B, dd, *J* = 132, 53 Hz), −36.7 (1B, d, *J* = 140 Hz). IR (film, cm$^{-1}$): 2936 ($\nu_{C-H}$), 2897 ($\nu_{C-H}$), 2885 ($\nu_{C-H}$), 2837 ($\nu_{C-H}$), 2537 (br, $\nu_{B-H}$), 1444, 1419, 1364, 1229. ESI HRMS: *m/z* for C$_6$H$_{20}$B$_9$OS: calcd. *m/z* 238.2119 [M]$^-$, obsd. *m/z* 238.2123 [M]$^-$.

### 4.6. General Procedure for the Synthesis of Complexes **5–9**

To a solution of **3** in the dry THF under argon atmosphere, the 3-fold excess of potassium *tert*-butoxide was added. The mixture was stirred for 15 min at ambient temperature and ~10 mol.% excess of nickel(II) phosphine complex was added. The reaction mixture immediately changed its color to different shades of green. The reaction mixture was stirred for 30 min at ambient temperature and the solution was evaporated under reduced pressure. The purification of crude product was carried out by column chromatography in dichloromethane as an eluent.

*3,3-dppe-1-MeO(CH$_2$)$_2$S-closo-3,1,2-NiC$_2$B$_9$H$_{10}$* (**5**). The synthesis was carried out using **3** (0.17 g, 0.48 mmol), *t*-BuOK (0.16 g, 1.43 mmol), and [(dppe)NiCl$_2$] (0.28 g, 0.53 mmol) in THF (20 mL) to give green solid of **5** (0.22 g, 68% yield). $^1$H NMR (acetone-d$_6$, ppm): $\delta$ 7.92 (4H, m, P*Ph*$_2$), 7.85 (4H, m, P*Ph*$_2$), 7.54 $\div$ 7.41 (12H, br m, P*Ph*$_2$), 3.14 (3H, s, OC*H*$_3$), 3.01 (1H, m, SCH$_2$C*H*$_2$OCH$_3$), 2.89 (1H, m, SCH$_2$C*H*$_2$OCH$_3$), 2.87 $\div$ 2.81 (4H, br m, -PC*H*$_2$C*H*$_2$P-), 2.57 $\div$ 2.39 (2H, br m, SC*H*$_2$CH$_2$OCH$_3$), 1.90 (1H, s, C*H*$_{carb}$), 4.4 $\div$ 0.6 (9H, m, B*H*). $^{13}$C NMR (acetone-d$_6$, ppm): $\delta$ 134.19 (d, *J* = 4.7 Hz, Ph), 134.14 (d, *J* = 4.7 Hz, Ph), 132.98 (d, *J* = 4.7 Hz, Ph), 132.93 (d, *J* = 4.7 Hz, Ph), 131.0 (*p*-Ph), 130.9 (*p*-Ph), 128.72 (d, *J* = 5.1 Hz, Ph), 128.67 (d, *J* = 5.1 Hz, Ph), 128.24 (d, *J* = 5.1 Hz, Ph), 128.19 (d, *J* = 5.1 Hz, Ph), 71.1 (S*C*H$_2$CH$_2$OCH$_3$), 57.6 (O*C*H$_3$), 56.4 (*C*S$_{carb}$), 52.1 (*C*H$_{carb}$), 36.1 (S*C*H$_2$CH$_2$OCH$_3$), 29.3 (d, *J* = 23.5 Hz, -P*C*H$_2$CH$_2$P-). $^{11}$B NMR (acetone-d$_6$, ppm): $\delta$ 5.5 (1B, d, *J* = 119 Hz), −7.2 (2B, d, *J* = 140 Hz), −11.3 (1B, d, *J* = 162 Hz), −12.8 (2B, d, *J* = 184 Hz), −16.8 (2B, d, *J* = 150 Hz), −20.8 (1B, d, *J* = 128 Hz). $^{31}$P NMR (acetone-d$_6$, ppm): $\delta$ 59.3 (dppe). IR (film, cm$^{-1}$): 3064 ($\nu_{C-H}$), 2989 ($\nu_{C-H}$), 2932 ($\nu_{C-H}$), 2898 ($\nu_{C-H}$), 2830 ($\nu_{C-H}$), 2561 (br, $\nu_{B-H}$), 2539 (br, $\nu_{B-H}$), 1591, 1572, 1488, 1439, 1416, 1382, 1314, 1251. ESI HRMS: *m/z* for C$_{31}$H$_{41}$B$_9$NiOP$_2$S: calcd. *m/z* 718.2237 [M+K]$^+$, obsd. *m/z* 718.2238 [M+K]$^+$. UV (acetone, nm): $\lambda$ 339, 431, 589, 711.

*3,3-(Me$_2$PhP)$_2$-1-MeO(CH$_2$)$_2$S-closo-3,1,2-NiC$_2$B$_9$H$_{10}$* (**6**). The synthesis was carried out using **3** (0.17 g, 0.48 mmol), *t*-BuOK (0.16 g, 1.43 mmol), and [(Me$_2$PhP)$_2$NiCl$_2$] (0.22 g, 0.53 mmol) in THF (20 mL) to give a green solid of **6** (0.16 g, 59% yield). $^1$H NMR (acetone-

d$_6$, ppm): $\delta$ 7.68 (2H, m, P*Ph*), 7.48 (8H, m, P*Ph*), 3.56 ÷ 3.48 (2H, br m, SC*H*$_2$CH$_2$OCH$_3$), 3.32 (3H, m, OC*H*$_3$), 3.07 ÷ 2.98 (2H, br m, SCH$_2$C*H*$_2$OCH$_3$), 1.70 (3H, d, *J* = 8.1 Hz, PC*H*$_3$), 1.61 (1H, d, *J* = 10.8 Hz, C*H*$_{carb}$), 1.51 (3H, d, *J* = 8.1 Hz, PC*H*$_3$), 1.36 (3H, d, *J* = 4.4 Hz, PC*H*$_3$), 1.29 (3H, d, *J* = 8.9 Hz, PC*H*$_3$), 3.1 ÷ 0.6 (9H, m, B*H*). $^{13}$C NMR (acetone-d$_6$, ppm): $\delta$ 130.7 (Ph), 130.4 (Ph), 129.9 (Ph), 128.4 (Ph), 71.3 (SCH$_2$CH$_2$OCH$_3$), 57.5 (OCH$_3$), 53.8 (CS$_{carb}$), 44.1 (d, *J* = 19.1Hz, CH$_{carb}$), 35.9 (SCH$_2$CH$_2$OCH$_3$), 17.5 (d, *J* = 35.9 Hz, PCH$_3$), 15.3 (d, *J* = 23.8 Hz, PCH$_3$), 13.4 (d, *J* = 22.7 Hz, PCH$_3$), 12.9 (d, *J* = 24.8 Hz, PCH$_3$). $^{11}$B NMR (acetone-d$_6$, ppm): $\delta$ −5.0 (2B, d, *J* = 130 Hz), −8.7 (1B, d, *J* = 142 Hz), −11.2 (1B, d, *J* = 167 Hz), −12.8 (1B, d, *J* = 168 Hz), −15.4 (2B, d, *J* = 132 Hz), −20.4 (1B, d, *J* = 152 Hz) −22.4, (1B, d, *J* = 156 Hz). $^{31}$P NMR (acetone-d$_6$, ppm): $\delta$ 6.9 (d, $J_{PP}$ = 48.5 Hz), −11.8 (d, $J_{PP}$ = 48.5 Hz). IR (film, cm$^{-1}$): 3094 ($\nu_{C-H}$), 3067 ($\nu_{C-H}$), 2988 ($\nu_{C-H}$), 2927 ($\nu_{C-H}$), 2834 ($\nu_{C-H}$), 2547 (br, $\nu_{B-H}$), 1482, 1438, 1421, 1379, 1302, 1284, 1249. ESI HRMS: *m/z* for C$_{21}$H$_{39}$B$_9$NiOP$_2$S: calcd. *m/z* 596.2075 [M+K]$^+$, obsd. *m/z* 596.2077 [M+K]$^+$. UV (acetone, nm): $\lambda$ 342, 433, 589, 705.

*3,3-(MePh$_2$P)$_2$-1-MeO(CH$_2$)$_2$S-closo-3,1,2-NiC$_2$B$_9$H$_{10}$* (**7**). The synthesis was carried out using **3** (0.25 g, 0.70 mmol), *t*-BuOK (0.24 g, 2.10 mmol), and [(MePh$_2$P)$_2$NiCl$_2$] (0.41 g, 0.77 mmol) in THF (20 mL) to give a green solid of **7** (0.30 g, 63% yield). $^1$H NMR (acetone-d$_6$, ppm): $\delta$ 7.81 ÷ 7.70 (6H, br m, P*Ph*), 7.46 ÷ 7.23 (14H, br m, P*Ph*), 3.57 ÷ 3.49 (1H, br m, SC*H*$_2$CH$_2$OCH$_3$), 3.49 ÷ 3.42 (1H, br m, SCH$_2$C*H*$_2$OCH$_3$), 3.34 (3H, m, OC*H*$_3$), 2.96 (2H, t, *J* = 6.7 Hz, SCH$_2$CH$_2$OCH$_3$), 1.93 (3H, d, *J* = 9.8 Hz, PC*H*$_3$), 1.55 (3H, d, *J* = 8.1 Hz, PC*H*$_3$), 0.82 (1H, d, *J* = 14.3 Hz, C*H*$_{carb}$), 3.6 ÷ 0.5 (9H, m, B*H*). $^{13}$C NMR (acetone-d$_6$, ppm): $\delta$ 133.5 (d, *J* = 8.5 Hz, Ph), 133.4 (d, *J* = 47.5 Hz, *ipso*-Ph), 133.2 (d, *J* = 7.1 Hz, Ph), 132.9 (d, *J* = 9.7 Hz, Ph), 132.1 (d, *J* = 8.9 Hz, Ph), 132.0 (d, *J* = 46.2 Hz, *ipso*-Ph), 130.6 (d, *J* = 8.6 Hz, Ph), 130.5 (d, *J* = 8.6 Hz, Ph), 130.0 (Ph), 128.4 (Ph), 128.3 (d, *J* = 9.7 Hz, Ph), 128.1 (d, *J* = 9.3 Hz, Ph), 71.6 (SCH$_2$CH$_2$OCH$_3$), 57.8 (OCH$_3$), 51.8 (CS$_{carb}$), 47.2 (d, *J* = 23.6 Hz, CH$_{carb}$), 36.2 (SCH$_2$CH$_2$OCH$_3$), 20.2 (d, *J* = 39.5 Hz, PCH$_3$), 10.0 (d, *J* = 33.6 Hz, PCH$_3$). $^{11}$B NMR (acetone-d$_6$, ppm): $\delta$ −3.8 (2B, d, *J* = 130 Hz), −7.5 (1B, d, *J* = 142 Hz), −10.5 (1B, d, *J* = 163 Hz), −12.2 (1B, d, *J* = 167 Hz), −13.6 (1B, d, *J* = 164 Hz), −14.6 (1B, d, *J* = 137 Hz), −18.7 (1B, d, *J* = 126 Hz), −21.5, (1B, d, *J* = 131 Hz). $^{31}$P NMR (acetone-d$_6$, ppm): $\delta$ 15.7 (d, $J_{PP}$ = 43.1 Hz), 1.5 (d, $J_{PP}$ = 43.1 Hz). IR (film, cm$^{-1}$): 3067 ($\nu_{C-H}$), 2997 ($\nu_{C-H}$), 2935 ($\nu_{C-H}$), 2893 ($\nu_{C-H}$), 2830 ($\nu_{C-H}$), 2553 (br, $\nu_{B-H}$), 1487, 1439, 1383, 1320, 1291, 1250. ESI HRMS: *m/z* for C$_{31}$H$_{43}$B$_9$NiOP$_2$S: calcd. *m/z* 699.3095 [M+NH$_4$]$^+$, obsd. *m/z* 699.3099 [M+ NH$_4$]$^+$. UV (acetone, nm): $\lambda$ 345, 445, 589, 738.

*3,3-(EtPh$_2$P)$_2$-1-MeO(CH$_2$)$_2$S-closo-3,1,2-NiC$_2$B$_9$H$_{10}$* (**8**). The synthesis was carried out using **3** (0.25 g, 0.70 mmol), *t*-BuOK (0.24 g, 2.10 mmol), and [(EtPh$_2$P)$_2$NiCl$_2$] (0.43 g, 0.77 mmol) in THF (20 mL) to give a green solid of **8** (0.24 g, 48% yield). $^1$H NMR (CD$_3$CN, ppm): $\delta$ 8.11 (2H, br m, P*Ph*), 7.71 (7H, br m, P*Ph*), 7.55 (11H, br m, P*Ph*), 3.24 (2H, m, SC*H*$_2$CH$_2$OCH$_3$), 3.24 (3H, s, OC*H*$_3$), 2.13 (2H, m, SCH$_2$C*H*$_2$OCH$_3$), 2.55–2.25 (4H, br m, PC*H*$_2$CH$_3$), 1.54 (1H, C*H*$_{carb}$), 1.08 (3H, m, PCH$_2$C*H*$_3$), 0.96 (3H, m, PCH$_2$C*H*$_3$), 3.5–0.4 (9H, m, B*H*). $^{13}$C NMR (CD$_3$CN, ppm): $\delta$ 134.2 (Ph), 134.1 (Ph), 132.7 (Ph), 131.7 (Ph), 131.1 (Ph), 130.7 (Ph), 130.5 (Ph), 128.8 (Ph), 128.6 (Ph), 128.5 (Ph), 71.2 (SCH$_2$CH$_2$OCH$_3$), 58.1 (OCH$_3$), 38.6 (SCH$_2$CH$_2$OCH$_3$), 21.4 (PCH$_2$CH$_3$), 8.2 (PCH$_2$CH$_3$), 4.9 (PCH$_2$CH$_3$). $^{11}$B NMR (CD$_3$CN, ppm): $\delta$ −5.1 (1B, d, *J* = 136 Hz), −8.7 (2B, d, *J* = 129 Hz), −12.0 (1B, d, *J* = 188 Hz), −13.5 (2B, d, *J* = 159 Hz), −15.9 (1B, d, *J* = 144 Hz), −17.7 (1B, d, *J* = 179 Hz), −22.8 (1B, d, *J* = 149 Hz). $^{31}$P NMR (acetone-d$_6$, ppm): $\delta$ 21.8 (d, $J_{PP}$ = 35.3 Hz), 12.6 (d, $J_{PP}$ = 35.3 Hz). IR (film, cm$^{-1}$): 2933 ($\nu_{C-H}$), 2881 ($\nu_{C-H}$), 2853 ($\nu_{C-H}$), 2544 (br, $\nu_{B-H}$), 1487, 1455, 1439, 1368. ESI HRMS: *m/z* for C$_{33}$H$_{47}$B$_9$NiOP$_2$S: calcd. *m/z* 732.2962 [M+Na]$^+$, obsd. *m/z* 732.2958 [M+ Na]$^+$. UV (acetone, nm): $\lambda$ 345, 481, 601, 701.

*3,3-(Bu$_3$P)$_2$-1-MeO(CH$_2$)$_2$S-closo-3,1,2-NiC$_2$B$_9$H$_{10}$* (**9**). The synthesis was carried us-ing **3** (0.25 g, 0.70 mmol), *t*-BuOK (0.24 g, 2.10 mmol), and [(Bu$_3$P)$_2$NiCl$_2$] (0.41 g, 0.77 mmol) in THF (20 mL) to give a reddish solid of **9** (0.20 g, 42% yield). $^1$H NMR (acetone-d$_6$, ppm): $\delta$ 3.48 (2H, SC*H*$_2$CH$_2$OCH$_3$), 3.26 (3H, OC*H*$_3$), 3.00 (2H, SCH$_2$C*H*$_2$OCH$_3$), 1.83 (12H, PC*H*$_2$CH$_2$CH$_2$CH$_3$), 1.78 (1H, C*H*$_{carb}$), 1.69 (12H, PCH$_2$C*H*$_2$CH$_2$CH$_3$), 1.44 (12H, PCH$_2$CH$_2$C*H*$_2$CH$_3$), 0.94 (18H, PCH$_2$CH$_2$CH$_2$C*H*$_3$), 3.6–0.5 (9H, m, B*H*). $^{13}$C NMR (acetone-

$d_6$, ppm): $\delta$ 71.5 (SCH$_2$CH$_2$OCH$_3$), 57.8 (OCH$_3$), 50.7 (CS$_{carb}$), 41.0 (d, $J$ = 15.0 Hz, CH$_{carb}$), 35.9 (SCH$_2$CH$_2$OCH$_3$), 25.9 (d, $J$ = 20.4 Hz, PCH$_2$CH$_2$CH$_2$CH$_3$), 24.3 (d, $J$ = 11.0 Hz, PCH$_2$CH$_2$CH$_2$CH$_3$), 24.2 (d, $J$ = 10.9 Hz, PCH$_2$CH$_2$CH$_2$CH$_3$), 13.3 (PCH$_2$CH$_2$CH$_2$CH$_3$). $^{11}$B NMR (acetone-$d_6$, ppm): $\delta$ −3.0 (1B), −4.5 (2B, d, $J$ = 129 Hz), −8.5 (2B, d, $J$ = 127 Hz), −14.1 (2B), −15.3 (2B), −21.5 (1B). $^{31}$P NMR (acetone-$d_6$, ppm): $\delta$ 10.5 (d, $J_{PP}$ = 43.6 Hz), 1.9 (d, $J_{PP}$ = 43.6 Hz). IR (film, cm$^{-1}$): 2964 ($\nu_{C-H}$), 2876 ($\nu_{C-H}$), 2821 ($\nu_{C-H}$), 2590 (br, $\nu_{B-H}$), 2508 (br, $\nu_{B-H}$), 1473, 1384, 1247. ESI HRMS: $m/z$ for C$_{29}$H$_{71}$B$_9$NiOP$_2$S: calcd. $m/z$ 685.4945 [M]$^+$, obsd. $m/z$ 685.4934 [M]$^+$. UV (acetone, nm): $\lambda$ 344, 457, 593, 697.

### 4.7. Single Crystal X-ray Diffraction Study

Crystallographic data for **5**·0.5Me$_2$CO: C$_{31}$H$_{41}$B$_9$NiOSP$_2$ 0.5(C$_3$H$_6$O) are triclinic, space group *P*-1: $a$ = 12.3355(7) Å, $b$ = 12.7617(7) Å, $c$ = 23.3086(12) Å, $\alpha$ = 94.623(2)°, $\beta$ = 95.020(2)°, $\gamma$ = 94.286(2)°, $V$ = 3630.9(3) Å$^3$, Z = 4, $M$ = 708.67, $d_{cryst}$ = 1.296 g·cm$^{-3}$. $wR2$ = 0.0876 calculated on $F^2_{hkl}$ for all 14,295 independent reflections with $2\theta < 52.1°$, (GOF = 1.051, $R$ = 0.0377 calculated on $F_{hkl}$ for 11,059 reflections with $I > 2\sigma(I)$).

The CCDC number 2243695 contains the supplementary crystallographic data for this paper. These data can be obtained free of charge via www.ccdc.cam.ac.uk/data_request/cif.

**Supplementary Materials:** The following supporting information can be downloaded at: https://www.mdpi.com/article/10.3390/inorganics11030127/s1, The NMR spectra of compounds **1–9** and crystallographic data on compound **5**.

**Author Contributions:** Conceptualization, M.Y.S.; methodology, M.Y.S. and I.B.S.; validation, D.K.S., M.Y.S. and I.B.S.; formal analysis, D.K.S., K.Y.S. and M.Y.S.; investigation, D.K.S.; data curation, I.B.S.; writing—original draft preparation, M.Y.S.; writing—review and editing, I.B.S.; supervision, I.B.S. All authors have read and agreed to the published version of the manuscript.

**Funding:** This research was supported by the Russian Science Foundation (21-73-10199).

**Institutional Review Board Statement:** Not applicable.

**Informed Consent Statement:** Not applicable.

**Data Availability Statement:** Not applicable.

**Acknowledgments:** The NMR, UV–Vis, and IR spectra as well as the single crystal X-ray diffraction data were obtained using equipment from the Center for Molecular Structure Studies at the A.N. Nesmeyanov Institute of Organoelement Compounds, operating with financial support from the Ministry of Science and Higher Education of the Russian Federation.

**Conflicts of Interest:** The authors declare no conflict of interest.

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
