# Peer review of "Half-Sandwich Nickelacarboranes Derived from [7-(MeO(CH2)2S)-7,8-C2B9H11]"

_inorganics, doi:10.3390/inorganics11030127_

Round 1

Reviewer 1 Report

The paper deals with the preparation of new 12-vertex nickellacarboranes that are substituted at one of the carbon atoms with a thioether group, which has the potential to act as an exo-polyhedral ligand. The compounds are characterized by NMR and mass spectroscopy, as well as X-ray diffraction analysis.

The reported nickellacarboranes have the potential to act as catalysts, and, in this regards, I recommend the authors to explore further their reactivity and catalytic activity.

Author Response

Thank you very much for your interest in our paper and its positive evaluation. Yes, of course, we plan to study the catalytic activity of the synthesized nickelacarboranes and their analogues with other substituents at the carbon atoms of the carborane ligand.

Reviewer 2 Report

The authors describe the first preparation of half-sandwich nickelacarboranes exploiting a straightforward synthetic route that chemists of the "boron cluster can use". The manuscript is well written; the abstract, introduction, results, and discussion are clear. However, I would suggest that the Authors would expand the conclusions by discussing the possible applications of their novel cages.

Author Response

Thank you very much for your interest in our paper and its positive evaluation. We expanded the Conclusion to give an idea of the possible applications of the synthesized new metallacarboranes.

Reviewer 3 Report

1.       It is better to modify the title of the manuscript as “Synthesis and characterization of the Half-Sandwich Nickelacarboranes Derived from [7-(MeO(CH2)2S)-7,8-C2B9H11]-“.

2.       Keywords should be five.

3.       Kindly form the molecular overlay plot in order to show the difference between molecules present in the asymmetric unit of complex 5.

4.       Kindly explore the intermolecular interactions that are weaker than Hydrogen bonding as such interactions also play an important role in deciding the properties of the crystal.

5.       In order to make the manuscript interesting for the readers, I highly recommend to explore the intermolecular interactions by Hirshfeld surface analysis. The analysis can be done by using Crystal Explorer software that is free to download and use for the academic purpose. For help, see the following

https://doi.org/10.1039/B818330A; http://dx.doi.org/10.1039/B704980C

https://doi.org/10.1039/D2RA07681K; https://doi.org/10.1016/j.rechem.2022.100600

6.       Kindly provide a table of the experimental details of the Single crystal XRD of complex 5.

7.       Kindly provide complete CIF and checkcif report for the review.

8.       Perform Cambridge structural data base search and compare the crystal structure of the complex 5 with the literature. 

Author Response

1. It is better to modify the title of the manuscript as “Synthesis and characterization of the Half-Sandwich Nickelacarboranes Derived from [7-(MeO(CH2)2S)-7,8-C2B9H11]-“.

Our manuscript is a standard synthetic work. The goal was to synthesize compounds that may be of interest from the viewpoint of their catalytic and biological activity. Any synthetic article implies characterization of newly obtained compounds. Therefore, it is not necessary to include this word in the Title, which should be as concise as possible.

2. Keywords should be five.

According to the Instructions for Authors, “three to ten pertinent keywords need to be added after the abstract”. In our opinion, it is the seven keywords we have used that best reflect the content of the manuscript. However, thanks for once again turning our attention to this point, as a result of which we decided to slightly change the order of the keywords.

3. Kindly form the molecular overlay plot in order to show the difference between molecules present in the asymmetric unit of complex 5.

We have prepared the Figure demonstrating the superimposition of ligands in two independent molecules of complex 5. However, since it contains information which is very close to Figure 3, we decided to place it in the Supplementary Materials.

4-5.       Kindly explore the intermolecular interactions that are weaker than Hydrogen bonding as such interactions also play an important role in deciding the properties of the crystal.

In order to make the manuscript interesting for the readers, I highly recommend to explore the intermolecular interactions by Hirshfeld surface analysis. The analysis can be done by using Crystal Explorer software that is free to download and use for the academic purpose. For help, see the following

https://doi.org/10.1039/B818330A; http://dx.doi.org/10.1039/B704980C

https://doi.org/10.1039/D2RA07681K; https://doi.org/10.1016/j.rechem.2022.100600

We agree that detailed crystal packing analysis can (however, unfortunately, in some cases can not) provide valuable information for explanation of some properties in the solid state. We also familiar with the Crystal Explorer program package and use it if necessary (for instance, in our recent paper related to triiodo-substituted carboranes - Molecules, 2023, 28, 875). Moreover, detailed visual analysis of all intermolecular contacts together with an energy analysis can provide even more information than ordinary Hirshfeld surface.

Unfortunately, in many papers you can see beautiful figures of Hirschfeld surfaces without any comments and explanation of the purpose for which this was done.

As we wrote above, our manuscript concerns the synthesis of new compounds. In this case, any physical properties of the synthesized compounds are not considered in this manuscript. With this respect, a detailed description of the crystal packing in here seems meaningless. Any research must have a purpose. A detailed analysis of the crystal packing may be of interest (and in some cases even important) for some physical properties that are not considered in this work. We believe that a standard synthetic paper should not be overloaded with structural details, especially if they do not use for explain the issues considered in the paper. Moreover, we believe that we have described even more structural details than necessary, and perhaps Figure 4 should be moved to the Supplementary Materials.

6. Kindly provide a table of the experimental details of the Single crystal XRD of complex 5.

We included the Experimental Table into the revised version of the Supplementary Materials. In fact, it just duplicates data from CIF-file. We believe that a brief description given in the manuscript will suffice for chemists who are little familiar with X-ray diffraction experiments, while a crystallographer will prefer to look at a CIF file than at an experimental table.

7. Kindly provide complete CIF and checkcif report for the review.

Complete CIF-file was submitted to the Editorial Board and Reviewers upon our first submission. If you downloaded it from CCDC, it can be truncated. The checkcif file is provided in the revised version.

8. Perform Cambridge structural data base search and compare the crystal structure of the complex 5 with the literature. 

Both compounds, with which it would be interesting to compare compound 5, are already mentioned in our text. The revised version describes this comparison in more detail (lines 123-127).

Round 2

Reviewer 3 Report

Recommend acceptance in its present form.